

# Feeding traces attributable to juvenile *Tyrannosaurus rex* offer insight into ontogenetic dietary trends

Joseph E. Peterson and Karsen N. Daus

Department of Geology, University of Wisconsin Oshkosh, Oshkosh, WI, United States of America

## ABSTRACT

Theropod dinosaur feeding traces and tooth marks yield paleobiological and paleoecological implications for social interactions, feeding behaviors, and direct evidence of cannibalism and attempted predation. However, ascertaining the taxonomic origin of a tooth mark is largely dependent on both the known regional biostratigraphy and the ontogenetic stage of the taxon. Currently, most recorded theropod feeding traces and bite marks are attributed to adult theropods, whereas juvenile and subadult tooth marks have been rarely reported in the literature. Here we describe feeding traces attributable to a late-stage juvenile *Tyrannosaurus rex* on a caudal vertebra of a hadrosaurid dinosaur. The dimensions and spacing of the traces were compared to the dentition of *Tyrannosaurus rex* maxillae and dentaries of different ontogenetic stages. These comparisons reveal that the tooth marks present on the vertebra closely match the maxillary teeth of a late-stage juvenile *Tyrannosaurus rex* specimen histologically determined to be 11–12 years of age. These results demonstrate that late-stage juvenile and subadult tyrannosaurs were already utilizing the same large-bodied food sources as adults despite lacking the bone-crushing abilities of adults. Further identification of tyrannosaur feeding traces coupled with experimental studies of the biomechanics of tyrannosaur bite forces from younger ontogenetic stages may reveal dynamic dietary trends and ecological roles of *Tyrannosaurus rex* throughout ontogeny.

# INTRODUCTION

Bite marks and feeding traces attributable to theropods dinosaurs provide important insight on behavior, physiology, and paleobiology. Furthermore, bite and feeding traces on fossilized bone represents a valuable demonstration of paleoecology; the interaction between two organisms as preserved in both traces and body fossils. Bite marks and feeding traces are relatively common in the fossil record, and are widely reported for theropod dinosaurs. Such traces have provided evidence of gregariousness and social interactions (*Tanke & Currie, 1998*; *Bell & Currie, 2009*; *Peterson et al., 2009*; *Currie & Eberth, 2010*), feeding behaviors and bone utilization (*Erickson & Olson, 1996*; *Chure, Fiorillo & Jacobsen, 1998*; *Hone & Watabe, 2010*; *Hone & Rauhut, 2010*), direct evidence of attempted predation (*Carpenter, 1998*; *Happ, 2008*; *DePalma et al., 2013*), and cannibalism (*Longrich et al., 2010*; *McLain et al., 2018*).

Corresponding author
Joseph E. Peterson,
petersoj@uwosh.edu

Despite the abundant record of theropod tooth marks, ascertaining the origins of feeding traces and bite marks can be challenging; determining the species responsible for the marks and establishing whether tooth marks are the result of active predation or scavenging largely depends on the taphonomic setting of the skeletal elements, the presence of shed teeth, and the location of the traces on the specimen in question (*Hunt et al., 1994*; *Bell & Currie, 2009*; *Hone & Rauhut, 2010*). However, most recorded cases of theropod feeding or the presence of bite marks are attributed to adult theropods, leaving the presence of juvenile and subadult tooth marks largely absent from the literature and discussion.

Here we report on the presence of feeding traces on the caudal vertebra of a hadrosaurid dinosaur (BMR P2007.4.1, "Constantine"). Based on the shape and orientation of the traces, and the known fauna of the Hell Creek Formation, they are interpreted to be feeding traces of a large theropod dinosaur, such as *Tyrannosaurus rex* (*Erickson & Olson, 1996*; *Horner, Goodwin & Myhrvold, 2011*). By comparing the dimensions and spacing of the traces with the maxillae and dentaries of specimens of *Tyrannosaurus rex* of different ontogenetic stages, we interpret these tooth marks to be feeding traces from a juvenile *Tyrannosaurus rex* and discuss the insights the specimen provides for juvenile tyrannosaur feeding behavior.

## Geologic setting

Specimen BMR P2007.4.1 is a partial hadrosaurid skeleton collected from the Upper Cretaceous Hell Creek Formation of Carter County, southeastern Montana in the Powder River Basin (Fig. 1). This specimen was collected on public lands under BLM Permit # M96842-2007 issued to Northern Illinois University and is accessioned at the Burpee Museum of Natural History in Rockford, IL. Exact coordinates for the location are on file in the paleontology collections at the Burpee Museum of Natural History (BMR), where the specimen is reposited.

The collection locality is composed of a 4 m fine-grained, gray-tan lenticular sandstone within a larger surrounding blocky mudstone unit (Fig. 2). The sandstone lacks bedforms, resulting from either (a) rapid accumulation (resulting in a lack of sedimentary structures), or (b) sedimentary structures that were obliterated by later currents or bioturbation, and is rich in rounded and weathered microvertebrate remains. The site is stratigraphically positioned approximately 44 m above the underlying Fox Hills–Hell Creek contact and overlies 0.5 m of siderite, which sits above a 5 m blocky mudstone. Grains are subrounded to subangular. Microvertebrate and fragmented macrovertebrate fossils are abundant and heavily rounded and abraded (*Peterson, Scherer & Huffman, 2011*). The fine-grained composition suggests a channel-fill deposit, overlying a floodplain deposit (*Murphy, Hoganson & Johnson, 2002*; *Peterson, Scherer & Huffman, 2011*). The taphonomic distribution of the elements and their stratigraphic position suggests the skeleton was subaerially exposed on a floodplain for a considerable period of time prior to burial, allowing for weathering, disarticulation, and removal of many skeletal elements.

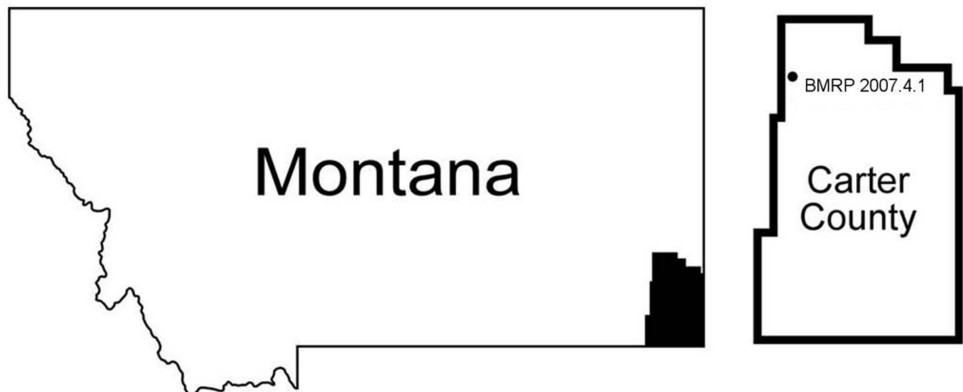

**Figure 1 Discovery location of BMR P2007.4.1.** Locality map showing the geographic location of specimen BMR P2007.4.1 in Carter County, Montana.

# MATERIALS & METHODS

Specimen BMR P2007.4.1 consists of weathered pelvic elements (sacrum, left and right ilia), three dorsal vertebrae, and two proximal caudal vertebrae (Fig. 3, Table 1). The dorsal vertebrae were too weathered for collection, though their dimensions and relative locations within the quarry assemblage were measured and documented. Additionally, a series of heavily-weathered bone fragments and a small shed theropod tooth (*Saurornithoides sp.*) were also collected.

The ilium of BMR P2007.4.1 possesses a number of hadrosaurid characters such as (1) a shallow morphology, (2) a ∼ 23° preacetabular process in medial view relative to the main body, and (3) a well-developed supra-acetabular process caudal to the acetabulum. While these characters are common among hadrosaurids, the stratigraphic position of BMR P2007.4.1 suggests it is attributable to the Late Cretaceous hadrosaurid *Edmontosaurus* (i.e., *Brett-Surman & Wagner, 2007*; *Campione, 2014*).

The centra of the two caudal vertebrae lack any evidence for hemal arch attachments, suggesting they are among the more cranial-positioned caudal vertebrae, such as C1–C4 (*Campione, 2014*). One of the caudal vertebrae possesses three v-shaped indentations on the ventral surface of the centrum (Figs. 4A–4E; Figs. S1 and S2). These traces feature collapsed cortical bone within the indentation, producing puncture marks (*sensu Binford, 1981*). The punctures penetrate 5 mm deep, are spaced 68 mm apart from their apical centers, show no signs of healing, and are inferred to have been created postmortem as feeding traces (e.g., *Noto, Main & Drumheller, 2012*; *Hone & Tanke, 2015*; *McLain et al., 2018*). The v-shape preserved in each puncture indicates that the original teeth would have possessed a prominent keel, though no striations from serration marks are present in the traces (*sensu D'Amore & Blumensehine, 2009*).

The large size and shape of the punctures suggests that they were produced by a large-to medium-bodied carnivore. Such carnivores from the Hell Creek Formation include tyrannosaurs such as *Tyrannosaurus rex* (*Erickson & Olson, 1996*; *Horner, Goodwin &*

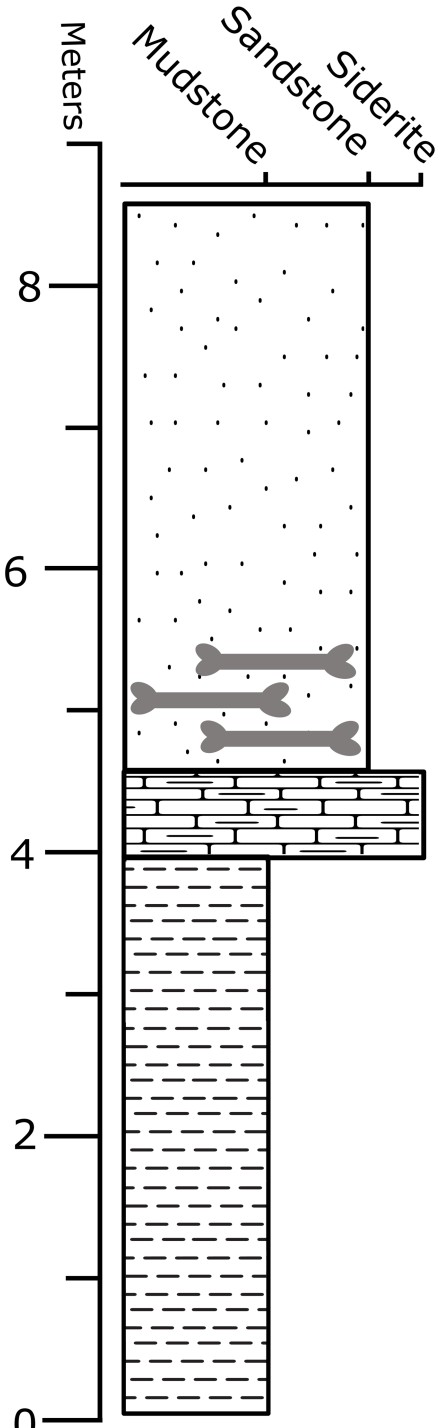

**Figure 2** **Stratigraphic column of the "Constantine" Quarry.** Stratigraphy of the BMR P2007.4.1 "Constantine" Quarry.

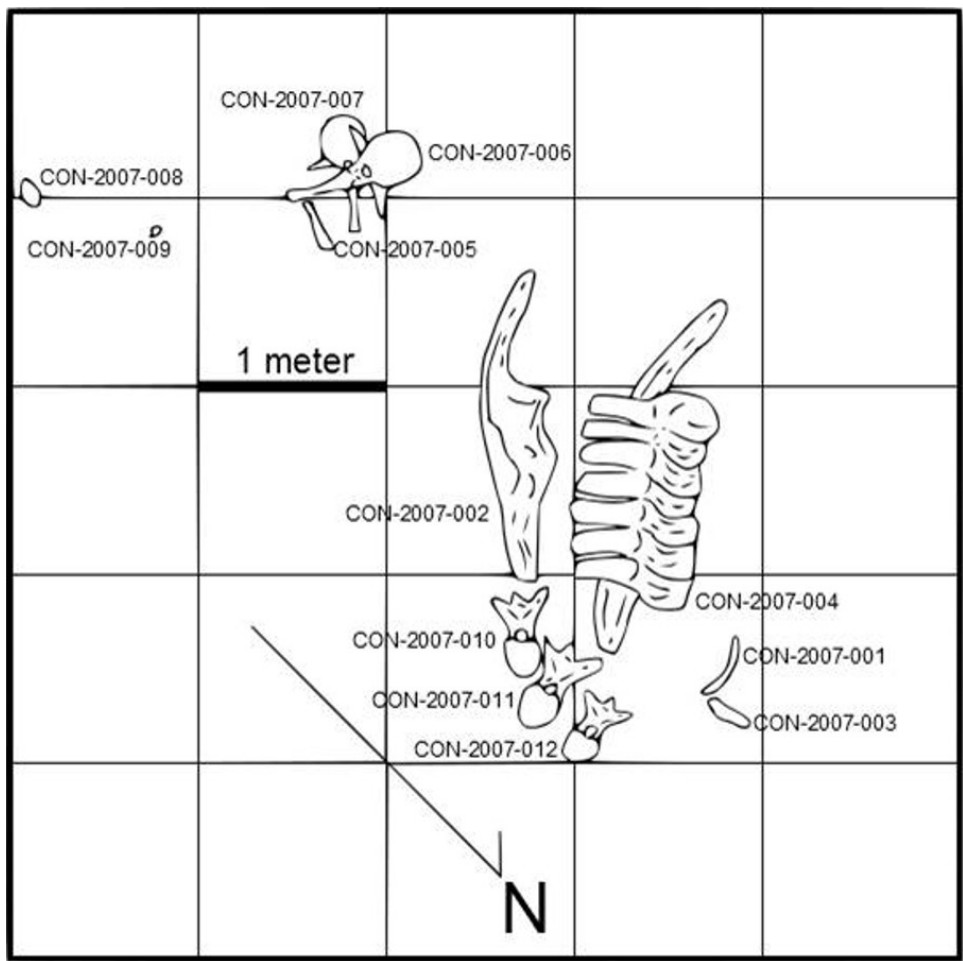

**Figure 3 Map of the BMR P2007.4.1 "Constantine" Quarry.** Dorsal vertebrae (field numbers CON-2007-010, CON-2007-011, and CON-2007-012) were too weathered for collection, though their relative locations were mapped. Note the relative association of dorsal and caudal vertebrae, and pelvic elements.

*Myhrvold, 2011*), medium-sized dromaeosaurids such as *Dakotaraptor steini* (*DePalma et al., 2015*), and crocodylians such as *Borealosuchus sternbergii*, *Brachychampsa montana*, and *Thoracosaurus neocesariensis* (*Matsumoto & Evans, 2010*). By comparing the shape and orientation of the traces to the teeth of these carnivores from the Hell Creek Formation, they are hypothesized to be bite marks from a large theropod dinosaur, such as *Tyrannosaurus rex* (*Erickson & Olson, 1996*); crocodylian teeth are circular in cross-section and too small, and dromaeosaurid teeth—even large dromaeosaurids such as *D. steini*—are too small and laterally-compressed to have produced the punctures observed on BMRP2007.4.1.

To test this hypothesis, the punctures on the caudal vertebra of BMR P2007.4.1 were first coated in Rebound[TM] 25 platinum-cure silicone rubber (Smooth-On) in order to make a silicone peel of the punctures in order to better visualize the morphology and dimensions of the teeth responsible for the traces (Figs. 5A–5B; Figs. S3 and S4). These "teeth" were then compared with the dental dimensions and spacing of two *Tyrannosaurus* maxillae and

**Table 1  Skeletal elements from BMR P2007.4.1.** Recovered and recorded skeletal elements from the "Constantine Quarry" (BMR P2007.4.1) and taphonomic condition.

| Field number | Element | State/Condition |
| --- | --- | --- |
| CON-2007-001 | Rib fragment | Abraded |
| CON-2007-002 | Left ilium | Heavily weathered |
| CON-2007-003 | Rib fragment | Abraded |
| CON-2007-004 | Sacrum and right ilium | Heavy to moderate weathering |
| CON-2007-005 | Neural arch | Fractured, but mild weathering |
| CON-2007-006 | Caudal vertebra | Mild weathering |
| CON-2007-007 | Caudal vertebra | Mild weathering |
| CON-2007-008 | Bone fragment | Heavily abraded |
| CON-2007-009 | Shed *Saurornithoides sp.* tooth | No apparent abrasion |
| CON-2007-010 | Dorsal vertebra | Heavily weathered, not collected |
| CON-2007-011 | Dorsal vertebra | Heavily weathered, not collected |
| CON-2007-012 | Dorsal vertebra | Heavily weathered, not collected |

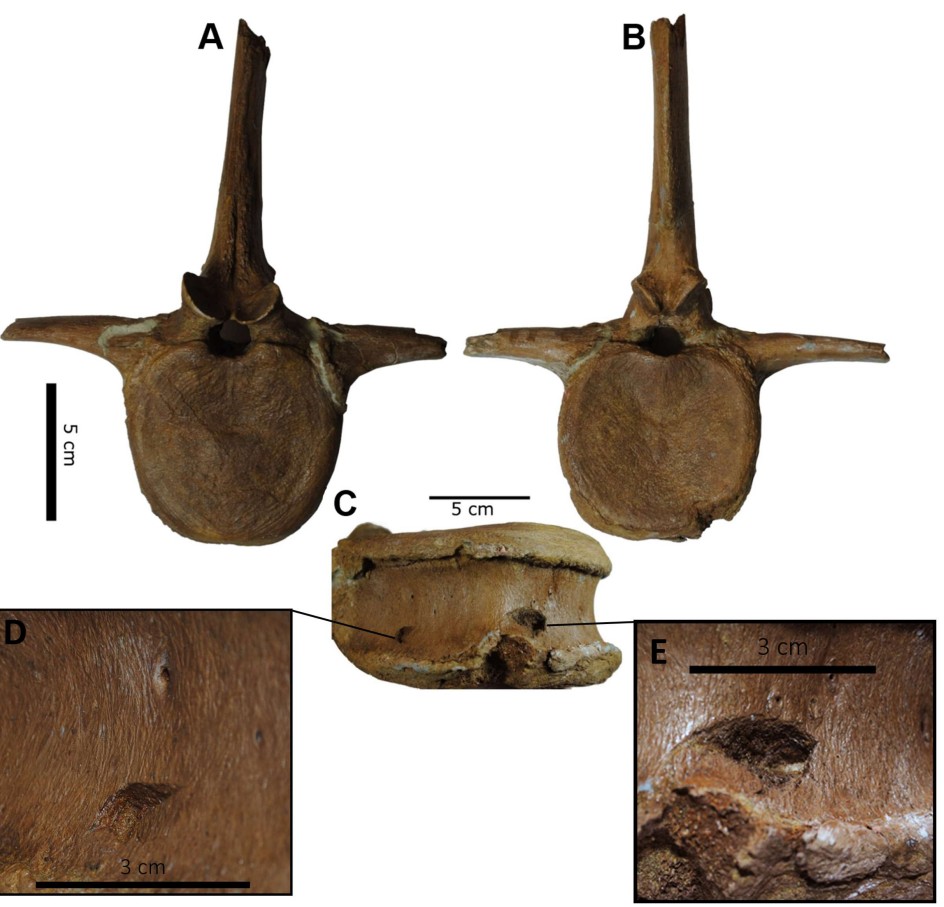

**Figure 4  Punctured caudal vertebra of BMR P2007.4.1.** BMR P2007.4.1 in anterior (A) posterior (B) and ventral (C), including the two elliptical punctures on the ventral surface of the centrum (D, E).

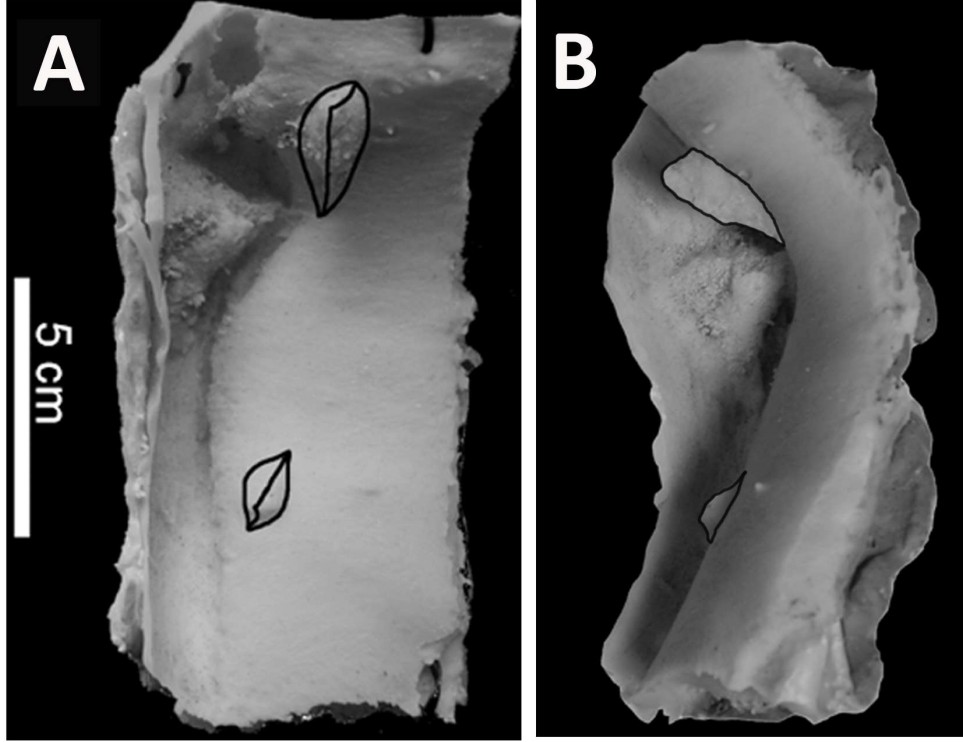

**Figure 5  Silicone peel produced from BMR P2007.4.1.** Silicone peel produced from the ventral surface of the punctured caudal vertebra of BMR P2007.4.1 in vertical (A), and lateral (B) views. Note the traced outlines demonstrating the shape of the tooth casts.

dentaries. To approximate the ontogenetic stage of the tyrannosaur, a late-stage juvenile specimen (BMR P2002.4.1, "Jane") histologically determined to be approximately 11–12 years old at the time of death (*Erickson et al., 2006*) that possesses laterally compressed, sharp crowns, and a mature specimen (BHI 3033, "Stan") with robust, blunt crowns were utilized.

All specimens were digitized via triangulated laser texture scanning with a NextEngine 3D Laser Scanner, capturing data at seven scanning divisions in high-definition (2.0k points/in 2). The resulting digital models were built with the NextEngine ScanStudio HD Pro version 2.02, and finalized as STL models (Figs. S1–S8). Scanning was conducted at the Department of Geology at the University of Wisconsin-Oshkosh in Oshkosh, WI.

The tooth spacing of both adult and late-stage juvenile tyrannosaur maxillae and dentaries were measured for both immediately-adjacent teeth and teeth from alternating replacement positions (i.e., *Zahnreihen*), and compared with the spacing of the punctures (Figs. 6A–6B), similar to *Fahlke*'s (*2012*) investigation of likely *Basilosaurus* bite marks on specimens the smaller whale *Dorudon*. Furthermore, the cross-sectional morphology of adult and late-stage juvenile tyrannosaur maxillae and dentaries were measured labiolingually and mesiodistally at a 5 mm apical depth for each tooth crown, and plotted with measurements from the punctures found on BMR P2007.4.1 (Fig. 7).

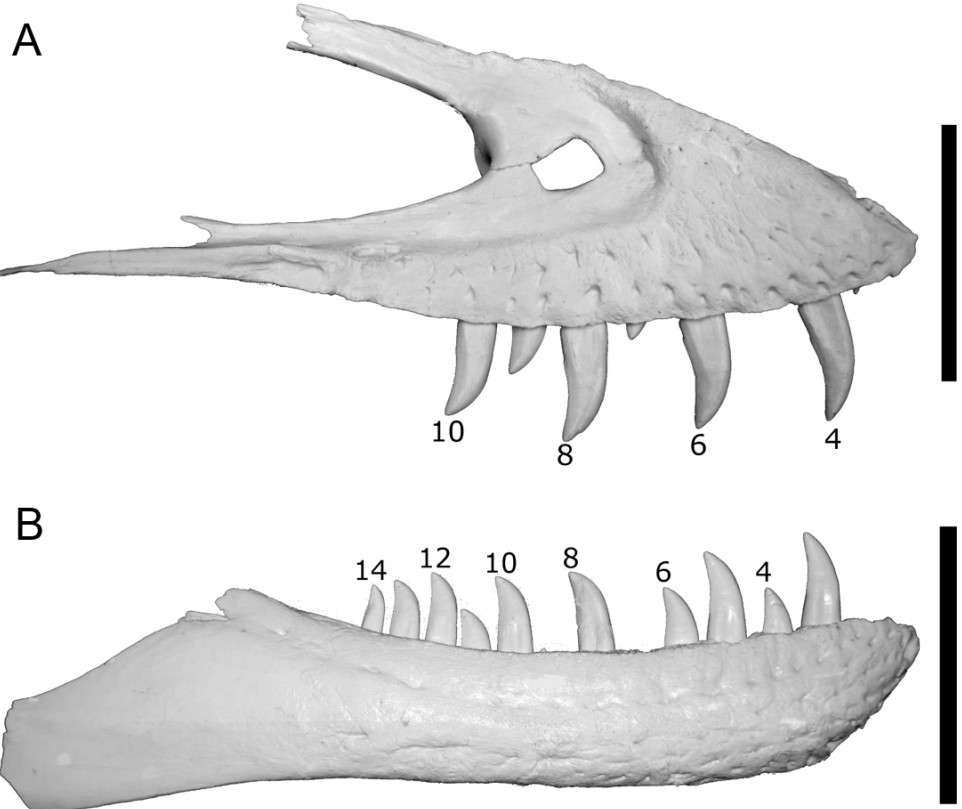

**Figure 6 Casts of BMR P2002.4.1 maxilla (A) and dentary (B) to illustrate the tooth positions used for spacing measurements.** Note the alternating replacement of teeth. Scale bars equal 10 cm.

## RESULTS

The mesiodistal width measurements from the silicone peel taken from BMR P2007.4.1 average 7.8 mm and the labiolingual depth average was 5.2 mm. Maxillary and dentary teeth of the adult *Tyrannosaurus* (BHI 3033) were found to be too large and widely spaced to have produced the punctures (Figs. 7, 8A and 8B; Figs. S4 and S5; Tables 2A–2C). For BHI 3033, the average dentary tooth crown mesiodistal width at 5 mm depth was 7.13 mm, and the average dentary tooth crown labiolingual depth at 5 mm was 4.10 mm. The average maxillary crown mesiodistal width at 5 mm were 7.72 mm, and the average maxillary crown labiolingual depth at 5 mm averaged to 4.21 mm.

However, the teeth of BMR P2002.4.1 produced similarly shaped punctures at 5 mm apical depth (Figs. 7 and 9; Figs. S7 and S8; Tables 2B–2C). The puncture measurements taken from the peel, BMR P2007.4.1 demonstrate a mesiodistal width and labiolingual depth consistent with the measurements taken from the maxillary and dentary teeth of the late-stage juvenile *Tyrannosaurus*. When plotted against the mesiodistal width and labiolingual depth of the maxillary teeth, measurements from the peel taken from BMR P2007.4.1 fall well within the cluster radius created by the late-stage juvenile *Tyrannosaurus*,

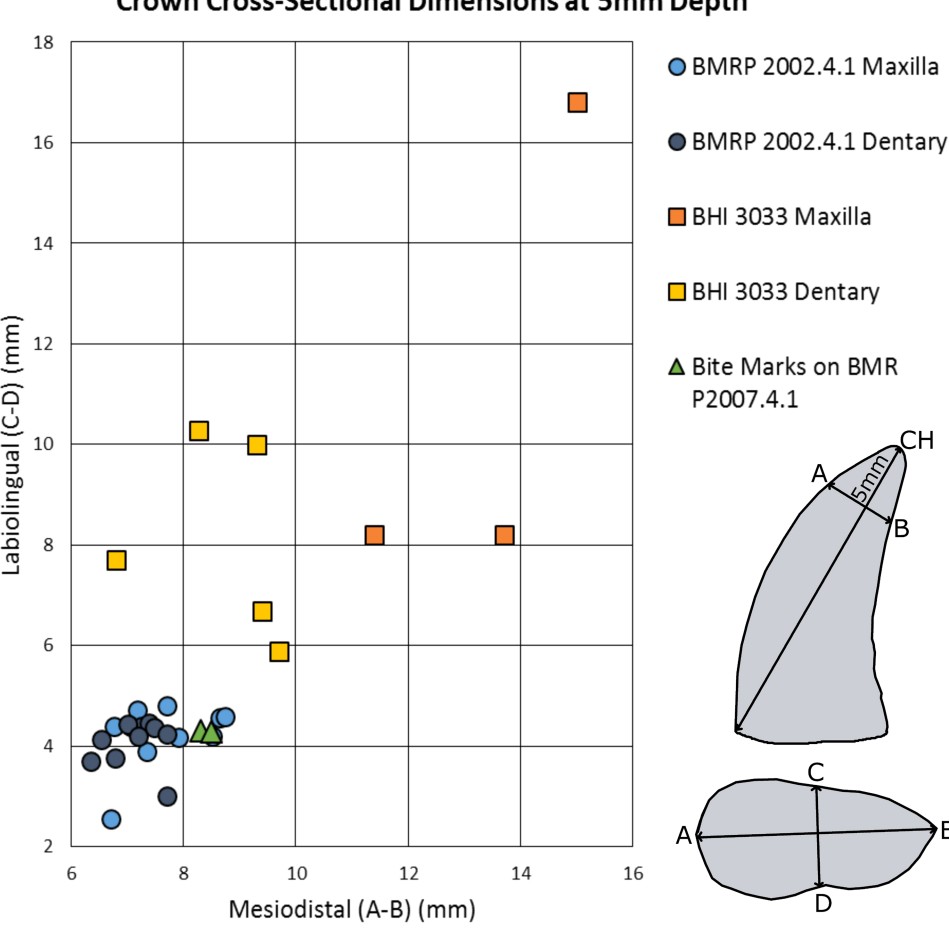

**Figure 7** Maxillary and dentary measurements for BMRP 2002.4.1 and BHI 3033 mesiodistal and labi-olingual dimensions at 5 mm depth compared to the bite marks on BMR P2007.4.1.

BMR P2002.4.1 (Fig. 7). Furthermore, the inferred crown spacing of the punctures closely matched those of the late-stage juvenile tyrannosaur maxilla (Tables 3A–3B).

# DISCUSSION AND CONCLUSIONS

While feeding traces and bite marks attributed to mature tyrannosaurids are well-documented in common Late Cretaceous taxa such as hadrosaurids and ceratopsians (i.e., *Fiorillo, 1991*; *Erickson et al., 1996*; *Erickson & Olson, 1996*; *Jacobsen, 1998*; *Farlow & Holtz, 2002*; *Fowler & Sullivan, 2006*; *Peterson et al., 2009*; *Bell & Currie, 2009*; *Fowler et al., 2012*; *DePalma et al., 2013*; *McLain et al., 2018*), the identification of juvenile tyrannosaur feeding traces adds insight into the role of juvenile theropods in Cretaceous ecosystems. The dimensions and spacing of the punctures closely matches the maxillary teeth of BMR P2002.4.1, a late-stage juvenile (11–12 yr old) tyrannosaur which incidentally itself possesses morphologically similar craniofacial lesions previously interpreted as a conspecific bite (*Peterson et al., 2009*).

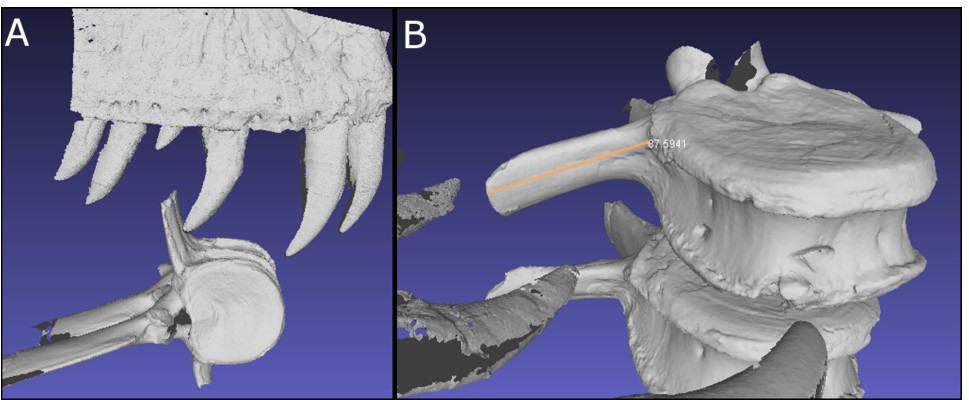

**Figure 8 Digitized comparisons between tyrannosaur maxillae and BMR P2007.4.1.** Interactive manipulation of digitized NextEngine 3D scan of a cast of the right maxilla of BHI #3033 and BMR P2007.4.1 caudal vertebra.

*Longrich et al. (2010)* reported on evidence of cannibalism in *T. rex* based on a number of bitten and scored remains of *Tyrannosaurus rex*, some attributed to juvenile or subadult individuals. However, many of these traces resemble the 'puncture and pull' bite marks that have previously been attributed to *T. rex* (*Erickson et al., 1996*; *Erickson & Olson, 1996*), and also include furrows and scores (*sensu Binford, 1981*).

Correlating traces in bone, such as tooth marks, to specific taxa and ontogenetic stages usually requires direct comparisons (e.g., *Peterson et al., 2009*; *Fahlke, 2012*). However, in cases where direct comparisons are not available, estimates can be made for tooth size, morphology, and spacing based on ontogenetic trajectories. While bite marks and feeding traces attributable to younger juvenile and hatchling tyrannosaurs have not yet been identified, the punctures present on the caudal vertebra of BMR P2007.4.1 provide direct evidence that late-stage juvenile *Tyrannosaurus rex* such as BMR P2002.4.1 possessed—at least in part—a similar diet as adults.

While bite marks resulting from active predation cannot easily be distinguished from postmortem feeding traces, the ventral position of the punctures in the caudal centrum of BMR P2007.4.1 suggests that the feeding was taking place postmortem with the hadrosaur already on its side (*Chure, Fiorillo & Jacobsen, 1998*). The afflicted vertebra is from the cranial-most part of the tail. Observations of the feeding behaviors of carnivoran mammals and birds indicate that in most cases, consumption of the axial skeleton occurs after limbs and viscera have been consumed (e.g., *Hill, 1980*; *Haglund, 1997*; *Carson, Stefan & Powell, 2000*; *Behrensmeyer, Stayton & Chapman, 2003*). Hadrosaur tails had substantial muscles such as m. ilio-ischiocaudalis and m. caudofemoralis longus (*Persons & Currie, 2014*) that might be a target of early stage consumption. However, the ventral bite traces on BMR P2007.4.1 suggest that the tyrannosaur was feeding after the haemal complexes and most of the superficial hypaxial muscles and m. caudofemoralis longus had been removed. As such the punctures on BMR P2007.4.1 suggest later-stage carcass consumption and postmortem feeding behaviors.

**Table 2 Measurements of tooth crowns of tyrannosaur specimens.** Mesiodistal and labiolingual measurements of teeth at 5 mm depth from the crown apex for (A) BHI 3033, (B) BMR P2002.4.1, and (C) the inferred bite marks on BMR P2007.4.1. All measurements are in mm.

**A**

| BHI 3033 | Maxilla | | Dentary | |
|---|---|---|---|---|
| | Mesiodistal | Labiolingual | Mesiodistal | Labiolingual |
| | 15 | 16.8 | 9.3 | 10.0 |
| | 11.4 | 8.2 | 8.27 | 10.27 |
| | 13.7 | 8.2 | 6.8 | 7.7 |
| | | | 9.4 | 6.7 |
| | | | 9.7 | 5.9 |

**B**

| BMR P2002.4.1 | Maxilla | | Dentary | |
|---|---|---|---|---|
| | Mesiodistal | Labiolingual | Mesiodistal | Labiolingual |
| | 6.77 | 4.4 | 7.06 | 4.39 |
| | 7.18 | 4.73 | 6.54 | 4.14 |
| | 7.35 | 3.9 | 6.78 | 3.77 |
| | 8.64 | 4.57 | 7.25 | 4.39 |
| | 7.91 | 4.19 | 7.39 | 4.47 |
| | 7.7 | 4.8 | 7.48 | 4.37 |
| | 8.74 | 4.59 | 7.0 | 4.44 |
| | 8.52 | 4.21 | 7.7 | 4.24 |
| | 6.71 | 2.56 | 7.19 | 4.2 |
| | | | 6.34 | 3.7 |
| | | | 7.7 | 3.01 |

**C**

| BMR P2007.4.1 "Bite Marks" | Mesiodistal | Labiolingual |
|---|---|---|
| | 8.3 | 4.31 |
| | 8.5 | 4.3 |

The identification of penetrating bite marks attributable to not only *Tyrannosaurus rex*, but an individual of 11–12 years of age can potentially allow for the determination of the ontogeny of bite force in *Tyrannosaurus rex* and for comparison with other theropods (e.g., *Barrett & Rayfield, 2006*; *Gignac et al., 2010*; *Bates & Falkingham, 2012*). Studies on the estimated bite forces of an adult *Tyrannosaurus rex* have yielded a wide range of results. Estimates based on muscle volume proposed bite forces between 8,526 and 34,522 N, coupled with tooth pressures of 718–2,974 MPa, and a unique tooth morphology and arrangement to promote fine fragmentation of bone during osteophagy (*Gignac & Erickson, 2017*). However, estimates incorporating likely muscle fiber length produced results over 64,000 N for adult *T. rex* (*Bates & Falkingham, 2018*). Juvenile *T. rex* such as BMR P2002.4.1 have much narrower and blade-like tooth morphologies and were unlikely to have been able to withstand similar bite forces at this ontogenetic stage. *Bates & Falkingham (2012)* estimated a maximum bite force for BMR P2002.4.1 at 2,400–3,850

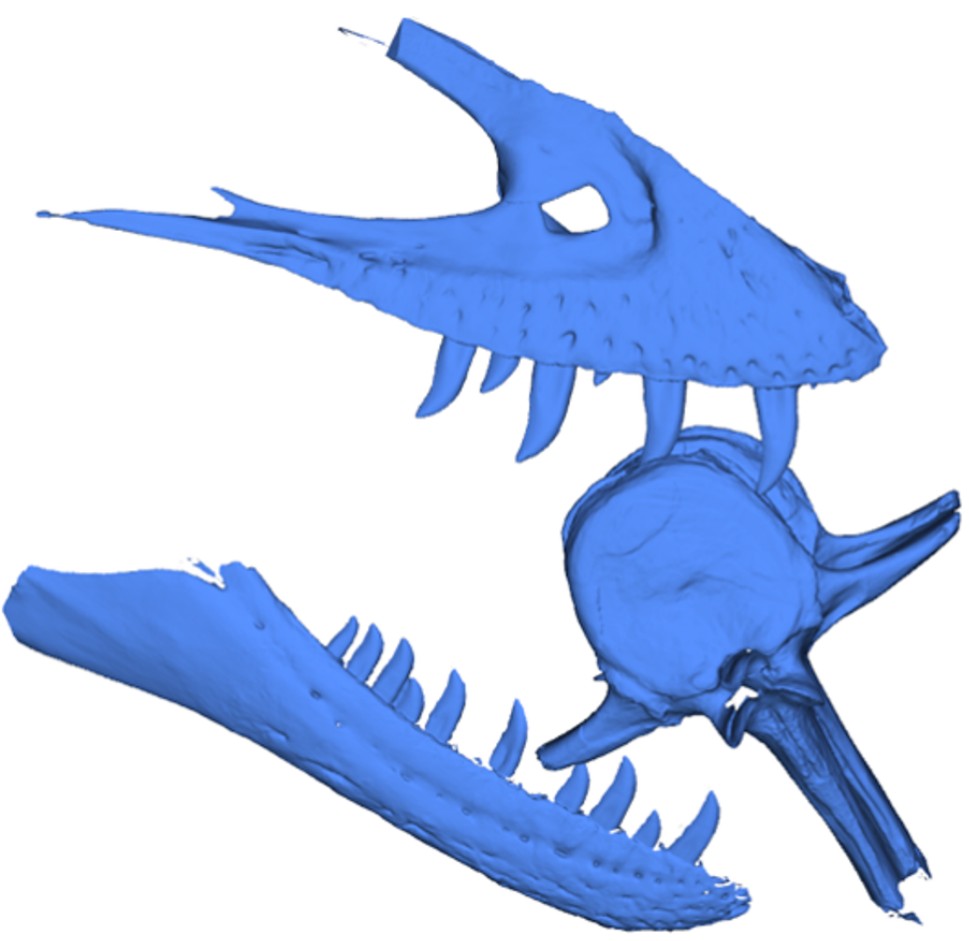

**Figure 9** **Digitized comparisons between BMR P2002.4.1 and BMR P2007.4.1.** Interactive manipulation of digitized NextEngine 3D scan of a cast of the right maxilla and dentary of BMR P2002.4.1, and BMR P2007.4.1 caudal vertebra.

N, and hypothesized that an increase in bite force during growth could indicate a change in feeding behavior and dietary partitioning while approaching adulthood.

Observations on extant crocodylians have documented a wide variety of dietary partitioning during ontogeny (e.g., *Tucker et al., 1996*; *Platt et al., 2006*; *Platt et al., 2013*). In the American crocodile (*Crocodylus acutus*), hatchling and small juveniles have a dietary overlap of over 80%, commonly feeding upon insects and crustaceans (*Platt et al., 2013*). Alternatively, larger juveniles, subadults, and adults possess a dietary overlap of over 75%, consisting of more birds, mammals, fish, and other reptiles (*Platt et al., 2013*). Comparable ontogenetic dietary partitions were also observed in Morelet's crocodile (*Crocodylus moreletii*) (*Platt et al., 2006*), and in Australian freshwater crocodiles (*Crocodylus johnstoni*) (*Tucker et al., 1996*). However, crocodylians are less discriminant of food sources when scavenging (e.g., *Antunes, 2017*). While the punctures present on BMR P2007.4.1 are likely from postmortem scavenging behaviors of a juvenile tyrannosaur, the degree of dietary overlap or partitioning between juvenile and adult tyrannosaurs remains unresolved.

**Table 3  Measurements of crown spacing in tyrannosaur specimens.** Tooth crown spacing between maxillary (A) and dentary (B) teeth in the juvenile tyrannosaur BMR P2002.4.1. All measurements are in mm.

**A**

| Crown spacing | Maxillary (mm) |
| --- | --- |
| **4–6** | 70.2 |
| **6–8** | 73.3 |
| **8–10** | 62.8 |
| **Average** | 68.7 |

**B**

| Crown spacing | Dentary(mm) |
| --- | --- |
| **4–6** | 53.3 |
| **6–8** | 49.8 |
| **8–10** | 39.2 |
| **10–12** | 33.7 |
| **12–14** | 33 |
| **Average** | 41.8 |

Despite not yet possessing the same feeding mechanisms of an adult *Tyrannosaurus rex* (i.e., bone-crushing and osteophagy), the punctures present on BMR P2007.4.1 demonstrate that late-stage juvenile and subadult tyrannosaurs were already biomechanically capable of puncturing bone during feeding, and were doing so without the large, blunt dental crowns of adults. Further identification of tyrannosaur feeding traces from different ontogenetic stages coupled with experimental studies of the biomechanics of tyrannosaur bite forces may reveal more insight into dynamic dietary trends and ecological role of *Tyrannosaurus rex* throughout ontogeny.

## Institutional Abbreviations

**BHI**      Black Hills Institute of Geologic Research, Hill City, SD, USA
**BMR**      Burpee Museum of Natural History, Rockford, IL, USA

## ACKNOWLEDGEMENTS

We wish to thank the 2007 Northern Illinois University field crew for assistance in the excavation of the BMR P2007.4.1, including Samuel Adams, Ryan Hayes, Erik Gulbrandsen, and David Vaccaro. We wish to offer particular appreciation to the Northern Illinois University students Christina Constantine-Laughlin and the late Dan Bocklund, who first discovered the specimen and to whom this study is dedicated. We also thank Kelsey Marie Kurz for assistance with specimen preparation. We thank Josh Matthews and Scott Williams of the Burpee Museum of Natural History for access to specimens, and Doug Melton of the Miles City, MT Bureau of Land Management office for assistance with permitting. We thank Jonathan Warnock for providing valuable feedback from early versions of the manuscript. We also that John R. Hutchinson for editing the manuscript, and Stephanie Drumheller-Horton and Eric Snively for offering constructive reviews. Finally, we are

grateful to the University of Wisconsin Oshkosh for recognition of this research at the 2018 UW Oshkosh Celebration of Scholarship Symposium.

### Funding
The authors received no funding for this work.

### Competing Interests
The authors declare there are no competing interests.

### Author Contributions
- Joseph E. Peterson conceived and designed the experiments, performed the experiments, analyzed the data, contributed reagents/materials/analysis tools, prepared figures and/or tables, authored or reviewed drafts of the paper, approved the final draft.
- Karsen N. Daus performed the experiments, analyzed the data, prepared figures and/or tables, authored or reviewed drafts of the paper, approved the final draft.

### Data Availability
Peterson, Joseph (2018): Supplemental Files for "Feeding traces attributable to juvenile Tyrannosaurus rex offer insight into ontogenetic dietary trends". figshare. Figure. https://doi.org/10.6084/m9.figshare.7424945.v2.

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
