# Peer review of "Feeding traces attributable to juvenile Tyrannosaurus rex offer insight into ontogenetic dietary trends"

_PeerJ, doi:10.7717/peerj.6573_

## Round 0.1 · original submission · Major Revisions

Two reviewers have checked the MS and made constructive criticisms, with one set being more in-depth and critical of some key issues in the paper, and both reviews urging the revised MS to be more explicit about how it would be ruled out that all other candidates except a juvenile T. rex left the bite marks (and R1's point about the Longrich study is apt). These points in particular, and all others raised, need to be addressed in the revised MS after moderate revisions, please; with Track Changes on one MS copy and a full Response to Reviewers covering all points. The reviewers agree there should a publishable paper here after such changes. R1 at least would need to re-review the MS to ensure its quality has progressed to an acceptable level. We look forward to your resubmission.

·

Basic reporting

“Feeding traces attributable to juvenile Tyrannosaurus rex offer insights into ontogenetic dietary trends” describes bite marks on a hadrosaurid dinosaur and provides interpretations of these traces in a larger context of dietary niche partitioning (or lack thereof) in T. rex. This study is well written and contributes meaningfully to a growing body of theropod bite mark studies. Figures and tables present the data clearly and fully (Table 2 and Figure 7 are a little redundant, but having the raw data on hand is appreciated). There are a few parts of this paper which need to be strengthened before I can recommend publication.

1) The authors do cite Longrich and colleague’s study on tyrannosaur bite marks, but they do not discuss the paper in much detail. I found this particularly surprising, because Longrich’s study also interprets their bite marks as feeding traces made by a juvenile or subadult T. rex. These traces seem to come from differently aged/sized individuals, and a more complete comparison should be included, especially since the authors are claiming that these new traces are the first published example of juvenile T. rex bite.

Longrich, Nicholas R., et al. "Cannibalism in Tyrannosaurus rex." PloS one 5.10 (2010): e13419.

2) The authors use the word “puncture” to describe the bite marks. Under Binford’s bite mark classification scheme, “puncture” has a very specific meaning relative to pits, scores or furrows. It in unclear if the authors are using the term in this context. If they are, Binford needs to be cited. If they are not, the authors should consider using the classification scheme since it is fairly straightforward and also allows better comparison and uniformity of terminology between their research and other vertebrate bite mark studies.

Binford, Lewis R. Bones: ancient men and modern myths. Academic Press, 2014.

See also D’Amore and Blumenschine, 2009 (full citation below) or the following for an abbreviated explanation of the classification scheme:

Drumheller, Stephanie K., and Christopher A. Brochu. "Phylogenetic taphonomy: a statistical and phylogenetic approach for exploring taphonomic patterns in the fossil record using crocodylians." Palaios 31.10 (2016): 463-478.

A few, minor edits:

Line 43 and elsewhere: I believe that the L in McLain (as in McLain et al., 2018) should be capitalized.

Line 54: Do the authors mean “fauna” rather than “phyla?” While I suppose it’s not technically wrong, using phyla in this context seems oddly broad.

Line 96: I believe there is a formatting hiccup here, with the text reading 23o instead of 23 degrees.

Experimental design

The underlying structure and reasoning of this project are sound, but there are a few areas that need further explanation to add detail and clarity.

1) What other predators are known from the locality? How were they excluded as potential trace makers? I am not necessarily arguing that the authors are incorrect in their identification, I only request that they flesh out their reasoning on how they came to the conclusion that these marks represent juvenile T. rex traces. As written, it sounds like the traces were only ever compared to tyrannosaur dentition. If all other taxa from the locality were too small to make these traces, list them and say so. If other groups’ teeth are morphologically distinct enough to exclude, say that as well. Simple association is a fairly weak argument for associating traces and trace makers, so adding more of this type of discussion would bolster the authors’ argument. The tooth shape and spacing measurements are compelling, but the omission of the rest needs to be addressed.

Along those lines, do any of the tooth marks preserve striations? Serrated teeth do not always leave striations behind, but when they are present, they can be very useful. If so, reference and add discussions addressing the following two papers on how to interpret striated marks and estimate minimum body size from them. This would also bolster the authors’ interpretations.

D'Amore, Domenic C., and Robert J. Blumensehine. "Komodo monitor (Varanus komodoensis) feeding behavior and dental function reflected through tooth marks on bone surfaces, and the application to ziphodont paleobiology." Paleobiology 35.4 (2009): 525-552.

D'Amore, Domenic C., and Robert J. Blumenschine. "Using striated tooth marks on bone to predict body size in theropod dinosaurs: a model based on feeding observations of Varanus komodoensis, the Komodo monitor." Paleobiology 38.1 (2012): 79-100.

If striations are not present, that is also worth mentioning.

2) There are some sections of this paper that I wish had just a bit more detail. For example, on line 43, the authors refer to an anatomical structure as “hadrosaur-like.” As a non-hadrosaur worker, this doesn’t actually tell me much. What is hadrosaur-like about it? Are there apomorphies that can be listed here? On line 105, the authors state that they compare the mark shape and orientation. To what? Were taxa other than T. rex observed in this study? Were the marks compared to other traces in the literature? As written, the authors don’t explain how they arrived at identifying the trace maker as T. rex.

3) The authors suggest that the presence of bite marks on caudal vertebrae suggest early access to the remains. The economy of bones does often guide the order of consumption (higher economy bones, such as femora, support more soft tissue, and are therefore targeted first while low economy bones, like phalanges, have less nutritional value and are therefore targeted last). This results in a predictable "scavenging sequence," in which certain regions of the body are typically consumed before others. Examples:

Hill, Andrew P., and A. K. Behrensmeyer. "Early postmortem damage to the remains of some contemporary East African mammals." Fossils in the making: Vertebrate Taphonomy and Paleoecology (1980): 131-152.

Haglund, William D. "Dogs and coyotes: postmortem involvement with human remains." Forensic taphonomy: the postmortem fate of human remains (1997): 367-381.

This sequence was originally based on mammalian predator and prey species, and deviations from the scavenging sequence are sometimes used to help differentiate predator and prey types. Here is an example involving birds:

Behrensmeyer, Anna K., C. Tristan Stayton, and Ralph E. Chapman. "Taphonomy and ecology of modern avifaunal remains from Amboseli Park, Kenya." Paleobiology 29.1 (2003): 52-70.

As another example, in which bears do seem to target the axial skeleton earlier than other predator groups:

Carson, E. Ann, Vincent H. Stefan, and Joseph F. Powell. "Skeletal manifestations of bear scavenging." Journal of forensic sciences 45.3 (2000): 515-526.

In most groups, viscera and limbs are targeted first while the axial skeleton is targeted last. This would not support the author’s interpretation that bite marks on the centra of cervical vertebrae would suggest early access to remains. Instead, this seems more consistent with late access or at least more complete consumption of the remains (a predator can work through all of the scavenging stages depending on the size of the prey and the duration of access to the remains). Since bite marks are only known on these vertebrae, this pattern would be more consistent with scavenging in the framework of the known scavenging sequence.

Validity of the findings

In the previous section, I discussed why I am not entirely convinced that these traces can be associated with predation over scavenging based on mark location. These two feeding strategies are notoriously difficult to differentiate in the fossil record. In this case, identifying these traces as predation rather than scavenging has bearing on the conclusions the authors draw regarding niche partitioning. If juvenile T. rex could in fact take larger prey, this would suggest less niche partitioning than expected given their relative size. The authors cite several examples using modern crocodylians as examples of groups with significantly different diets across ontogeny.

However, if these marks represent scavenging, the body size of the prey item has little bearing on niche partitioning during predation. To use modern crocodylians as an example again, some modern crocs have been known to scavenge adult elephants, even though they cannot successfully hunt these animals. This has been used to interpret some fossil crocodylian bite marks as incidents of scavenging:

Antunes, Miguel Telles. "Huge Miocene Crocodilians From Western Europe: Predation, Comparisons with the “False Gharial” and Size."

I think that the implications of both scavenging and predation should be discussed in this paper, because I do not think that the authors have adequately argued that the traces represent one behavior to the exclusion of the other.

Additional comments

Overall, I think this is a very interesting study. My suggestions are largely meant to strengthen the authors' existing arguments. I do feel that the addition of a section discussing the implications of scavenging rather than predation is warranted.

·

Basic reporting

The manuscript generally meets the criteria for publication. The description of the geological setting can use minor editing and wording changes. Additional citations will improve the discussion of T. rex bite force (Bates and Falkingham 2018), and indention of bone by ziphodont-toothed theropods (Gignac et al. 2010). Also cite Carpenter (1998) and Happ (2008) for records of predation attempts.

Experimental design

The methods are appropriate to the scope of the manuscript. Cite Fahlke (2012) as a study similar to this one.

Validity of the findings

The tooth marks appear to match a specimen of specific size and age. I suggest adding a sentence or two to the discussion about criteria you'd use to identify tooth marks of a tyrannosaur younger or older than one of this exact ontogenetic stage. How would you accommodate different tooth spacing and size?

Additional comments

The figures are clear, but the measurement template (Figure 7) is a dromeosaur tooth. A tyrannosaur tooth would be far more appropriate.

---

## Round 0.2 · Minor Revisions

The two reviewers have some mainly minor recommendations for revision; mostly wording but some deeper intellectual issues that need to be pondered and responded to while revising the MS. We look forward to the final draft!

·

Basic reporting

The authors have fully addressed all of my previous comments on their manuscript “Feeding traces attributable to juvenile Tyrannosaurus rex offer insights into ontogenetic dietary trends.” The updates have strengthened an already strong study. I only have a very small number of extremely minor comments. Line and page numbers in my comments are based off of the provided Word document with track changes.

Page 2, line 80: insert a space between 4 and m.

Page 3, line 112: caudalm should be caudal

Page 3, line 118: vertebra should be vertebrae

Page 3, line 125: (sensu D’Amore and Blumenschine, 2009) should probably be inserted here, regarding presence/absence of striations in bite marks left by ziphodont teeth. Full citation:

D'Amore, Domenic C., and Robert J. Blumensehine. "Komodo monitor (Varanus komodoensis) feeding behavior and dental function reflected through tooth marks on bone surfaces, and the application to ziphodont paleobiology." Paleobiology 35.4 (2009): 525-552.

Page 6, lines 236 and 244: Crocodylian (with a y) is technically the correct name for this clade.

Page 6, line 242: crocodile doesn’t need to be capitalized here.

Experimental design

No comment

Validity of the findings

No comment

·

Basic reporting

The reporting is sound. Another reference is suggested (see 3. Validity of the findings), and a few wording suggestions are on the edited manuscript. The references formatting is inconsistent.

Experimental design

Quite good, as before.

Validity of the findings

The ventrolateral position of the bite marks provides additional support for late-stage carcass consumption. Big Mesozoic dinosaurs had a lot of meat on the tail, unlike that available to modern scavengers on mammal and bird carcasses. That suggests the tail would be an early target. However, if this tyrannosaur was digging into the tail at an early stage from this direction, it would get a mouthful of haemal spine. I'd leave it there; otherwise you risk a speculative whirlpool of suggests, woulds, and ifs. (Do jaguars eat caimans' tails first? Their rostra aren't built to rummage in viscera. What about mammals scavenging on Nile crocs? Nile monitors feeding on Osteolemus? Shrikes on lizards? Komodo dragons go for guts first; crocs seem to eat whatever they want whole.)

Sentences addressing this issue are suggested, and a citation+reference added.

Additional comments

The authors circumspectly incorporate suggestions from both reviewers. They did not fully address the issue of identifying bite marks from a juvenile tyrannosaur whose tooth spacing does not exactly match the Burpee specimen. Happily, adding one word ("morphology"; line 207 of the edited manuscript) resolves the issue, especially with the added discussion of ontogenetic trajectory.

The tyrannosaur tooth template is welcome in Figure 7.

---

## Round 0.3 · accepted · Accept

I am satisfied by the revisions and am pleased to recommend your paper for publication! Well done.

#